# Engaging children in geosciences through storytelling and creative dance

Ana Matias[1], A. Rita Carrasco[1], Ana A. Ramos[2], Rita Borges[2]

[1] CIMA – Universidade do Algarve, 8000 Faro, Portugal

[2] Centro Ciência Viva de Tavira, 8800 Tavira, Portugal

*Correspondence to*: Ana Matias (ammatias@ualg.pt)

**Abstract.** Natural sciences have traditionally been disseminated in outreach activities as formal one-way presentations. Nevertheless, innovative strategies are being increasingly developed using arts, gamming, sketching, amongst others. This work aimed at testing an alternative and innovative way to engage non-expert audiences in ocean and coastal geology, through a combination of scientific concepts explanation with creative dancing. An informal education activity focusing on ocean dynamics was designed for 10-year-old students. It combines coastal science concepts (wind, waves, currents, and sand), storytelling techniques (narrative arc), and creative dance techniques (movement, imaginative play, and sensory engagement). A sequence of six exercises was proposed starting in the generation of offshore ocean waves and ending with sediment transport on the beach, during storm/fair-weather conditions. Scientific concepts were then translated into structured creative movements, within imaginary scenarios, and accompanied by sounds or music. The activity was performed six times summing 112 students. It was an inclusive activity given that all students in the class participated, including children with several mild types of cognitive and neurological impairment. The Science & Art activity aroused emotions of enjoyment and pleasure, and allowed an effective communication between scientists and school public. Moreover, the results provide evidence of the activity effectiveness to engage children and to develop their willingness to further participate in similar activities.

**Keywords**: coastal science; ocean literacy; storytelling; science engagement; geoscience communication; creative dance.

## 1. Introduction

The act of dissemination (and communication) is part and parcel of doing research. The main vehicle of scientific information relies within the scientific community, through peer-reviewed periodicals, generally focused on specific research areas and directed at well-circumscribed, specialized audiences (e.g., Gravina et al., 2017). Nevertheless, there is still a gap in the

effectiveness of such communication to the general public, with scientists often seen as being
trapped in the ivory tower (e.g. Baron, 2010) and commonly using scientific jargon hard to
understand by the common citizen. There are a vast range of approaches to engaging public
audiences with scientific concepts (Bultitude, 2011); Mesure (2007) identified over 1500 active
initiatives within the UK alone. There are three main forms of media used in science
communication to the general public: traditional journalism; live or face-to-face events, and
online interactions. According to Bultitude (2011), live events have the advantages of being
more personal, scientists are able to better control the content, engenders two-way
communication, and can involve partnering with other external organizations with
complementary expertise. The disadvantages are limited audience reach, resource intensive,
leading to low sustainability of activities, and can be criticised for only attracting audiences with
a pre-existing interest.
According to Kim (2012), effective communication of science lies in the processes of public
engagement with a problem or an issue relative to science; the processes of engagement
develops from the acts of exposing and focusing attention to the act of cognizing. Science
journalism and classroom instruction seem to hold strongly to the traditional learning-theory
paradigm that mere exposure to scientific knowledge would lead to scientific literacy and public
understanding (Kim, 2012). In this work, engagement will not be used in the same sense as
Public Engagement with Science, which has a specific meaning that refers to activities, events,
or interactions characterized by mutual learning among people of varied backgrounds, scientific
expertise, and life experiences who articulate and discuss their perspectives, ideas, knowledge,
and values in response to scientific questions or science-related controversies (McCallie et al.,
2009). Here, in terms of informal science education, engagement is a loosely defined term
referring to behaviours that demonstrate interest in, or interaction with science-related activity
or experience.
Recent work indicates that storytelling and narrative can help communicate science to non-
experts, within the wider context of "framing" as an important feature of public outreach
(Martinez-Conde and Macknik, 2017). Furthermore, strategies fusing arts and science (e.g.
using games, poetry, music, painting, sketching) are becoming a favoured medium for
conveying science to the public (e.g., Cachapuz (2014), Von Roten and Moeschler (2007),
Gabrys and Yusoff (2012)). Collaborative projects between artists and Science, Technology,
Engineering, and Mathematics (STEM) fields are not new, with renewed interest over the last
decades (Heras and Tàbara, 2014), hence Science, Technology, Engineering, Arts and
Mathematics - STEAM is increasingly replacing the traditional STEM designation. A maturing
body of work indicates that the arts can deeply engage people by focusing on the affective
domain of learning (i.e., engagement, attitude, or emotion) rather than on the cognitive domain
(i.e., understanding, comprehension, or application), which is often emphasized in science

education (Friedman, 2013). Therefore, science communication through art brings science to the public in ways that are engaging, instructive, artistic and, always, content-driven (Schwartz, 2014). Examples of "Science and Art" projects include theatre as a way of communicating coastal risk (Brown et al., 2017), hip-hop dance as a way of learning ecology (Wigfall, 2015), or art installations inspired in neuroscience laboratories (Lopes, 2015). Varelas et al. (2010) observed that while participating in a play representing STEM concepts, students engaged in understanding science from multiple perspectives. Embodied exercises situate abstract concepts in a concrete context, thus relating intangible ideas with corporeal information, and so rich multimodal distributed neural representations are forged (Hayes and Kraemer, 2017). Chang (2015) compiled an environmental science artwork database that consisted of 252 artworks, but only 4% included artistic mediums like poetry, dance and performances; the majority was from the visual arts domain. Good examples of STEM education through creative dance can be found in Landalf (1997) approaching earth sciences and in Abbott (2013) approaching mathematics. Creative dance is thus one mode for learning that involves using the body and the senses to gather information, communicate, and demonstrate conceptual understanding (Cone and Cone, 2012).

In Portugal, Afonso et al. (2013) reported that science teaching appeals to memorization of data and lacks abstract conceptual understanding. Geology education in particular is mostly associated to memorization (e.g. minerals and rocks), which drives students away from the geosciences. Moreover, science communication to the general public only occasionally covers geosciences, in comparison to other sciences such as astronomy, health, or biology, as can be deducted from an analysis of most newspapers records (consultation to the science section records of the Portuguese newspaper "Público"), although good examples can be found in science communication literature (e.g., Pedrozo-Acuña et al., 2019).

Coastal and marine geology have traditionally been disseminated in science outreach activities in the form of formal one-way presentations or, at best, field trips or lab experiences. The success of outreach actions and education programs requires knowing and understanding different audiences and strategizing how to reach them. So, efforts are kept now in the improvement of marine science literacy with accurate and appealing techniques that strengthen the learner's emotional connection to the ocean. The Intergovernmental Oceanographic Commission (IOC) of UNESCO stands that only through Ocean Literacy it will be possible to create an educated society capable of making informed decisions and caring for the preservation of Ocean's health (Santoro et al., 2017). In this context, effective geoscience communication activities addressing Principle 2 of Ocean literacy defined by the IOCommission: "The ocean and life in the ocean shape the features of the Earth" are in great need and aligned with UNESCO Sustainable Development Goal (SDG) 14: "Conserve and sustainably use the oceans, seas and marine resources for sustainable development", are in great need.

Aligned with SDG 14 and IOC Principle 2 of Ocean literacy, the objective of this work was to
develop an alternative and innovative activity to engage children in geosciences, by combining
scientific concepts transmission with creative dance. Moreover, this work intended to provide
additional arguments about the importance of arts (dance) and communication techniques
(storytelling) in engagement and effectiveness of geoscience programmes and develop their
willingness to participate in similar activities. Described activities were performed within the
framework of the outreach task of a research project devoted to the evolution and resilience of
barrier island systems (the EVREST project). EVREST project (more information in
https://evrest.cvtavira.pt/) identified natural and human processes that contributed to Ria
Formosa (south of Portugal) barrier island evolution (Kombiadou et al., 2019b) and developed a
framework to quantify barrier island resilience (Kombiadou et al., 2019a, 2018). The project,
led by a research centre (CIMA – Universidade do Algarve) also included Tavira Ciência Viva
Science Centre (devoted to disseminating science to the general public), the partner responsible
for facilitating the bridge between researchers and primary schools' students.

**2. Development of the activity "The Sea Rolls the Sand"**
An interdisciplinary activity was developed by merging techniques and tools from arts, science,
science communication and storytelling (Figure 1). The three main components were the
scientific content (the message to be communicated); the storytelling and metaphors (the verbal
way of communicating the message); and creative dance structure (the sensorial way of
communicating the message).

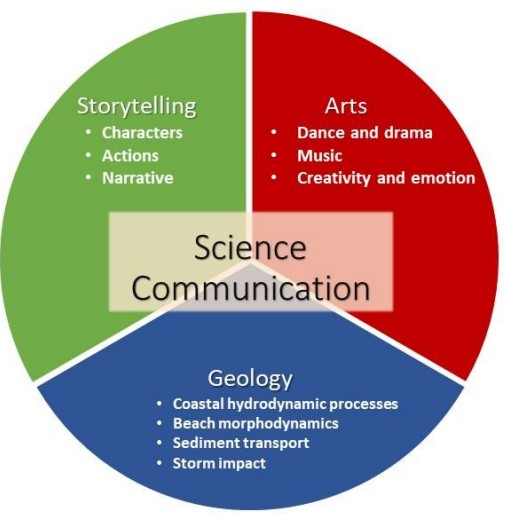


**Figure 1 - Scheme summarising the elements from each component to develop the interdisciplinary research.**


### 2.1. Scientific contents

The activity was developed to communicate concepts and processes related to marine and coastal morphodynamics to 10 years old students, attending the 4th grade. In Portugal, the geosciences are an academic discipline of the official primary school curricula. Nevertheless, geoscience contents are included in the generic discipline of "environmental studies", which includes basic knowledge of science such as the human body, solar system, monarchy history, earth surface morphology, water cycle, and protection of the environment. Within this discipline, there is a unit devoted to the sea – land interface.

The activity was composed by a series of six exercises (Figure 2) that were preceded by a simplified but accurate scientific explanation, adapted to the average expected pedagogical level, starting with an introduction, followed by basic geoscience concepts explanation, and enforcing the message with a resume at the end. The key geosciences concepts were wave, wave size, breaking waves, sand grain, sediment transport, beach dynamics, and seasonality. Waves form when the water surface is disturbed, for example, by wind, earthquakes or planetary gravitational forces. During such disturbances energy and momentum are transferred to the water mass and transmitted in the direction of the impelling force (e.g., Carter, 1988). At the shoreline, part of the incoming wave energy is reflected and is propagated back to the open sea, very much the way light bounces off a mirror; most of the incoming wave energy, however, is transformed to generate nearshore currents and sediment transport, and is ultimately the driving force behind morphological change at the coast (e.g., Masselink and Hughes, 2003). The portion of the coast most familiar to most people is the beach. The beach includes the adjacent seabed

bellow shallow marine waters, generally called the nearshore environment until the highest high
tide line. The beach is composed of nearly anything that can be transported by waves (e.g.,
Davis, 1996), predominantly sand but also gravel, mineral as well as organic, that come from
river discharge, cliff erosion, glacier melting, organic shells production, volcanic activity, and
ocean continental shelf, amongst others (e.g. Anthony, 2014). The exchange of beach sediment
between submerged and sub-aerial portions of the beach is accomplished by onshore-offshore
transport, mainly by waves, but aided sometimes by wind (e.g. Carter, 1988). Beach
morphology thus responds to changing wave conditions, and has a cyclic behaviour. In many
occasions, the cycles are seasonal; wave conditions during winter storms shifts sand offshore,
whilst calm conditions during the summer induce landward migration of sediments back to
upper parts of the beach (e.g., Komar, 1976).
Important associations from this activity are the connection between atmosphere, ocean and the
coast, and the insight between casual observations that the students make, i.e., their empirical
knowledge of the coast, for example, breaking waves, beach width, sand grains, and the science
behind it.
The scientific content was divided into three major hydrodynamic and morphodynamics
situations: wave generation and propagation, sediment transport and storm/fair-weather
conditions. Wind blowing on ocean surface and wave generation were explained not only to
elucidate how waves are generated but also to demonstrate the connection between separate
environments (atmosphere and the oceans). Wave propagation was used to illustrate energy
transference across the ocean surface, opposite to mass transference and to make the transition
from the ocean to the coastal environment, until waves break at the shore (Figure 3). The
generation of onshore currents under the presence of waves from the submerged to the sub-
aerial part of the beach was then introduced. Sediment transport by onshore currents was
explained as a straightforward effect, in the presence of grains in the bottom (lower block-
diagram and pink arrow on Figure 4). Here sediment variability, including shape, size and
composition, were introduced in relation to possible sources, such as volcanic rocks or coral
reefs.

| | Geology | Storytelling | Dance/movement | Example |
|---|---|---|---|---|
| 1 | Introduction to coastal geology | Exposition Action: preparing for the beach trip/applying sunscreen | Warmup | |
| 2 | Coastal & oceanic environments | Exposition Action: trip to the coast and dive into the ocean | Jumping Swimming movements | |
| 3 | Wind & wave generation Wave propagation | Rising action Action: making waves | Cadence Improvisation | |
| 4 | Wave induced currents Sediment transport | Rising action Action: currents moving grains, and breaking waves | Direction Improvisation Ball passage | |
| 5 | Storm waves Off/onshore currents Erosion/accretion | Climax Action: currents moving grains | Direction change Improvisation Ball passage | |
| 6 | Resume | Falling action Action: sunbathing | Relaxation | |


**Figure 2 - Activity outline: list of scenes (from 1 to 6), related scientific contents, associated storytelling moment and type of dance movements.**




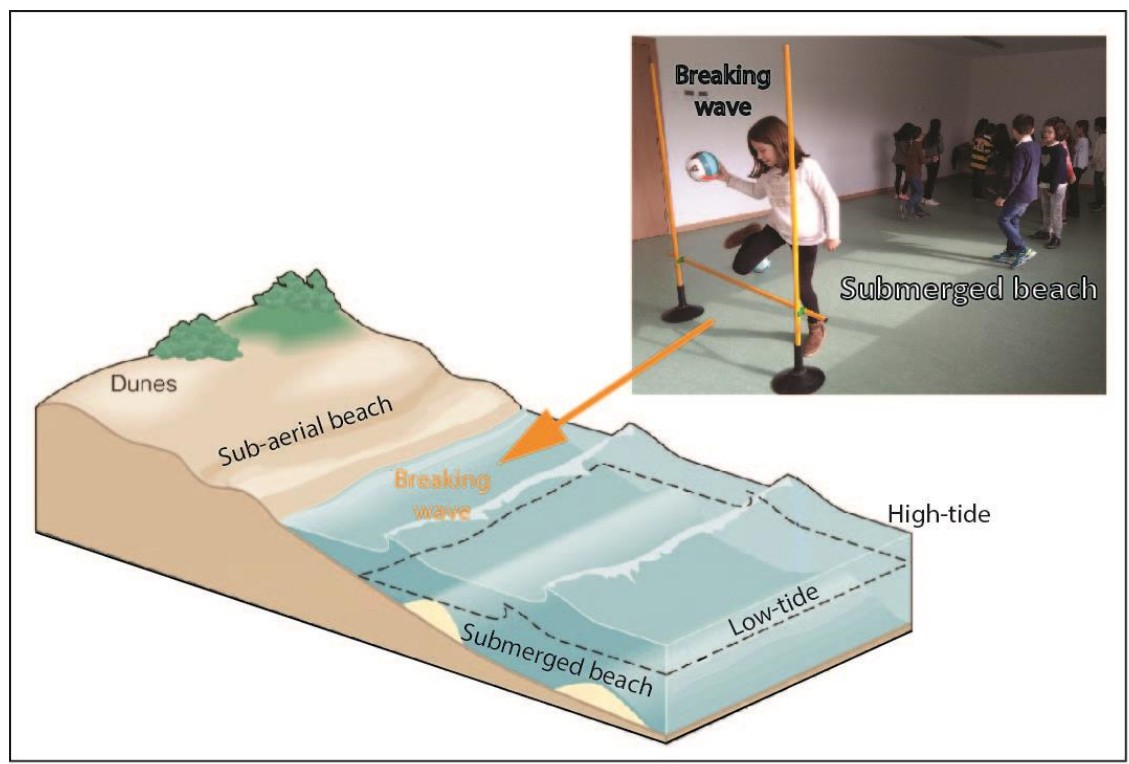

**Figure 3 - Coastal environments: dunes, sub-aerial beach and submerged beach. The photograph shows a**
**"breaking wave" with a jump over the yellow horizontal bar, representing the position that separates the sub-**
**aerial from the submerged beach (towards the right hand-side, where children are in two rows "propagating**
**waves").**

Wave height variations throughout the year were explained, by introducing the concept of storm
waves and induced sediment transport pattern (upper block-diagram and pink arrow on Figure
4). Because onshore currents generated by fair-weather were explained, offshore currents and
consequently beach erosion did not need an elaborated explanation. The alternation between
erosion and accretion, i.e., seasonality of waves and beach morphology depending on wave
height was reinforced, both as natural occurrences on a natural beach.

**2.2. Storytelling and metaphors**
As in any story, the activity had a theme, settings, scenes, characters, actions, and a narrative
arc. In broad terms, the narrative arc is the sequence of action shaped by the exposition, rising
action, crisis, climax and falling action (e.g. Hart, 2011). The theme of coastal dynamics is
immediately set in the introduction, when the scientific topic is addressed. The settings, i.e., the
natural environments, were built with psychomotricity equipment, but mostly appealing to
imagination. Psychomotricity is a holistic type of intervention by means of movement and play,
oriented towards humanism and respecting a child's development stage (cf., for example,
Vetter, 2019). It refers to psychomotor educational interventions (e.g., Perrotta, 2011) but also
to therapeutic practices (e.g., Ayres, 2005; Ingwersen et al., 2019), where there is a relation
between the psyche (mental processes) and motoric (physical activities). Typical psychomotor
equipment (cf., European Forum of Psychomotricity, 2016) for children includes colourful
hoops, balls, cones, mates, bags, blocks, and poles, that can be used isolated or as frames,
tunnels, tracks, climbing sets or balancing courses.
There were three main settings: the deep ocean, the beach under water and the sub-aerial beach
(Figure 3). The limit of the sub-aerial and submerged beach, i.e., the wave breaking position
was marked with two poles and a horizontal bar, while sediment balls of different sizes, colours,
shapes and textures represented sediments (Figure 3). The settings/scenario of the action
(marine and coastal environments) were also suggested by specific actions such as diving into
the ocean (jump over the horizontal bar), imaginary application of sunscreen, and sunbathing
(relaxation, Figure 4). Characters performed by students were beach users (scenes 1, 2 and 6,
Figure 2) and water particles (scenes 3 to 5, Figure 2).

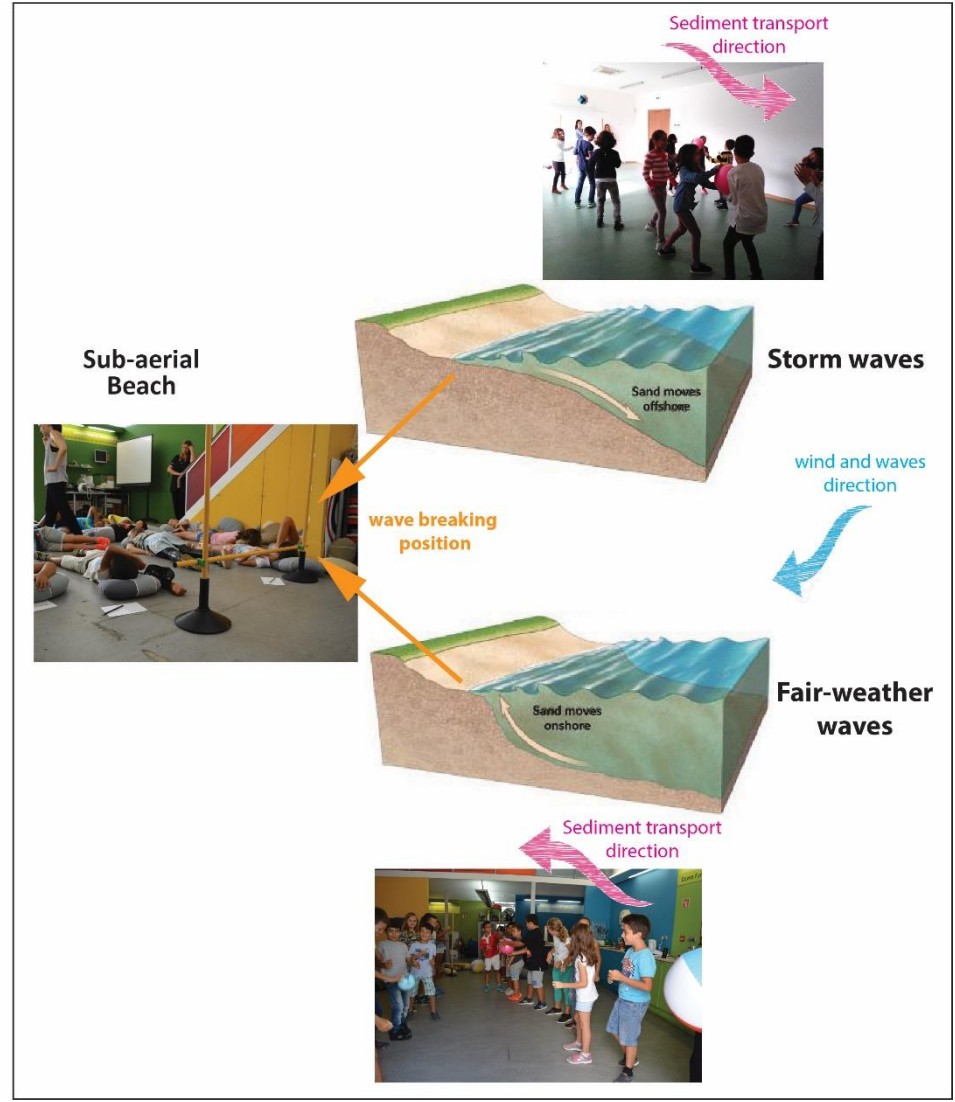


**Figure 4 - Coastal environments, coastal processes and metaphors. The image illustrates waves approaching**
**the coast, coming from the right side (blue arrow). The direction of sediment transport (pink arrows on top of**

The narrative consisted of a set of six practical actions (exercises) that were plotted in a predefined sequence of increasing complexity and excitement (at the beginning of the activity), with a sharp decline to relaxation (at the end of the activity), following the narrative arc (Figure 2). During scenes 1 and 2, an exposition to the theme and settings was conducted, obtained by the verbal explanation of the beach topic and by suggesting a sequence of actions that mimic a trip to the beach, finishing with the dive into the ocean; students (actors) embodied beach users. From scenes 3 to 4 settings were kept, but characters were changed, and actors embodied water particles, instead of beach users. The actions involved exercises of increasing complexity, reflecting a rise in action, as they impersonated water particles of the sea surface and then water particles as a current that transported grains to the shore. In scene 5, the climax was attained when storm waves reached the coast in several moments, and sediments could move in opposite directions. During scene 6, characters returned to beach users again; actors came out of the ocean and sunbathe, in a falling action (Figure 2 and 4).

**2.3 Creative dance structure**

According to Gilbert (2015), creative dance is a dance form that combines the mastery of movement with the artistry of expression. In creative dance, children generate, vary, and manipulate movement by using the elements of dance through the process of improvisation (Cone and Cone, 2012). The basic movement concepts used here derive from Laban Movement Analysis. Rudolf Laban's (1897-1958) philosophy was based on the belief that the human body and mind are one and inseparably fused ( e.g., Newlove and Dalby, 2004). It was Laban's firm belief that it is the birth right of every man to dance – not just trained dancers or folk dancers and the like, but all human beings (Newlove and Dalby, 2004). Laban Movement Analysis is a method to describe and analyse human movement and to establish a notation system with precision and clarity (cf., Laban, 1963). Laban's ideas have been picked up, reinterpreted, evolved and ramified, for example, to Dance Movement Psychotherapy (e.g., Best, 2008), programmes for individuals affected by complex needs (e.g., Price, 2008) and creative dance (e.g., Gilbert (2015). Structure and elements used here were also based in techniques described by several dance educators (Landalf, 1997; Carline, 2011; Cone and Cone, 2012; Abbott, 2013; Gilbert, 2015). The creative dance unit focused the effort concepts of time (fast/slow), space (direction), and flow (bond/free). A typical session of creative dance is composed of: 1)

warming up; 2) Exploring the concept; 3) Developing skills; 4) Creating; and 5) Cooling down
(Gilbert, 2015).
During the first exercise (scene 1), applying sunscreen, there was a warm up of muscles and
mobilization of articulations through light aerobic movements, such as bending, twisting and
curling (see dance/movement on Figure 2). During the second exercise (scene 2), students
jumped over the obstacle (diving into the sea, Figure 3), in turns, and made swimming free
movements across the space. In the third exercise (scene 3), students stand in two lines facing
each-other, consisted in reproducing several waves with the body curling up, with arms up, in a
cadence. The movement was repeated in a cadence of dance improvisation. During the fourth
exercise/scene, the two rows of students performed dance improvisation while passing different
balls (representing sediment transport) in the direction of the obstacle (the sub-aerial beach,
Figure 4), jumping to mimic breaking waves. In the fifth exercise/scene, students applied the
same type of movements than in the fourth exercise/scene, but listening a different soundtrack;
music changed in intensity and the balls moved to the obstacle when the music's intensity was
lighter and move in the opposite direction when the music was louder and more intense to
represent fair-weather waves and storm waves, respectively. During the sixth exercise/scene,
students spread thorough the available space and rested on the floor, while relaxing, and sensory
stimulation was induced by speech, appealing to sensations felt while sunbathing (sea smell,
warm on the skin, wind sensation, sand grains below the body).
Soundtracks included music/sounds with lyrics allusive to the sea (exercises 1, 2 and 6),
soundtracks of animation movies (exercise 2), sounds from nature (wind on exercise 3 and
waves on exercise 6), a Portuguese traditional theme (exercise 1), classical music (exercise 5),
and pop music (exercise 4). The activity was called "The Sea Rolls the Sand", which is the
name of a Portuguese traditional song. All musical themes had easy rhythmical and melody
compositions.

**3. "The Sea Rolls the Sand" activity implementation**

**3.1. Performing opportunities and institutional framework**
The activity was performed six times, within national and international initiatives. During the
first two times, the sessions were included in the activity of the "European Researcher Night",
in September, 29th, 2017. These sessions took place in the educational laboratory of the Tavira
Ciência Viva science centre, which was emptied as much as possible to create space for physical
activities. The other four sessions were included in a national initiative "Science and
Technology Week", in November 23th and 24$^{th}$, 2017. These sessions took place at three
(private and public) schools, in the classrooms and in the gym.
Overall 112 students participated in the activity, divided in school classes, varying between 15
and 22 students per session. Two classes in small schools in rural areas included students from
different grades; 1$^{st}$ and 4$^{th}$, in one case, 3$^{rd}$ and 4$^{th}$ in another case. Tavira municipality had 323
students attending 4$^{th}$ grade classes or mix classes, divided in 16 classes (with 13 to 26
students/class). Therefore, about 35% of all 4$^{th}$ grade students of the municipality participated in
the activity.
All students in the class participated, including children with cognitive impairment, attention
deficit disorder, amblyopia, light autism, hyperactivity and dyslexia.
Teachers assisted all sessions and had no intervention on the scientific topics or session
alignment; however, occasional teacher's interference occurred to assist behaviour control of the
class. In one of the sessions, a teacher assigned for cognitive impairment students was also
present, but no interference took place. There was no discussion or presentation in advance with
the teachers about the sessions' specific methods and contents. Teachers volunteered to
participate solely based on the information of the general topic. They were briefed about the
need of an empty room and that children should be wearing clothes appropriate for physical
activity.

**3.2. Activity evaluation by participants**
At the end of the activity, with children still laying over the room floor, small inquiries were
distributed to obtain an anonymous evaluation. Questions concerned: 1) if they enjoyed the
activity; 2) if they liked the movements; 3) if they liked the music; 4) how do they prefer to
learn science; 5) if they think they learnt something new; and 6) if they would like to repeat it,
and if so with another person or in another place.
From the 112 students that responded the inquiries, there was an even distribution of boys and
girls (51% were girls). Results showed that all children enjoyed themselves, and 80% enjoyed a
lot (Figure 5A). About 75% liked the movements a lot and only 1% was not sure about this.
Only one student did not like the music selection. After anonymously filling the inquiry, the
student stated: "I hate classical music".
According to the inquiry's responses, these children prefer to learn science through movement
and games, although field trips and laboratory experiments were also frequently selected
(20/112, Figure 5B). When questioned about how much they learned with the activity, 35%
answered they learned something new, and 60% answered they learned a lot, with 5% stating
they already knew everything. 99% of children want to repeat the activity, but 20% of the
students from one of the schools referred they preferred to do it elsewhere (Figure 5C).
The time constraints and the lack of personnel to assure children's supervision did not allow a
proper quantitative assessment of the schoolteacher's opinions. Nevertheless, teachers expressed
that "the activity was very nice and good for children this age". Additionally, some teachers
were concerned about some children's inability to follow entirely the scientific content, or not
having an appropriate behaviour all the time.
The researcher conducting and researchers assisting the activity observed that these children,
living in coastal areas, although having limited scientific background on coastal geology, have
plenty of empirical experience on the coast.

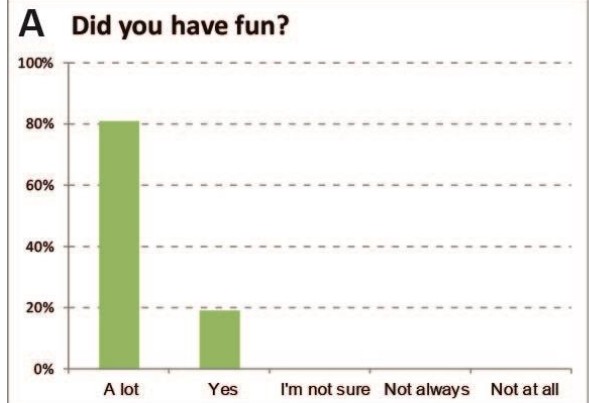

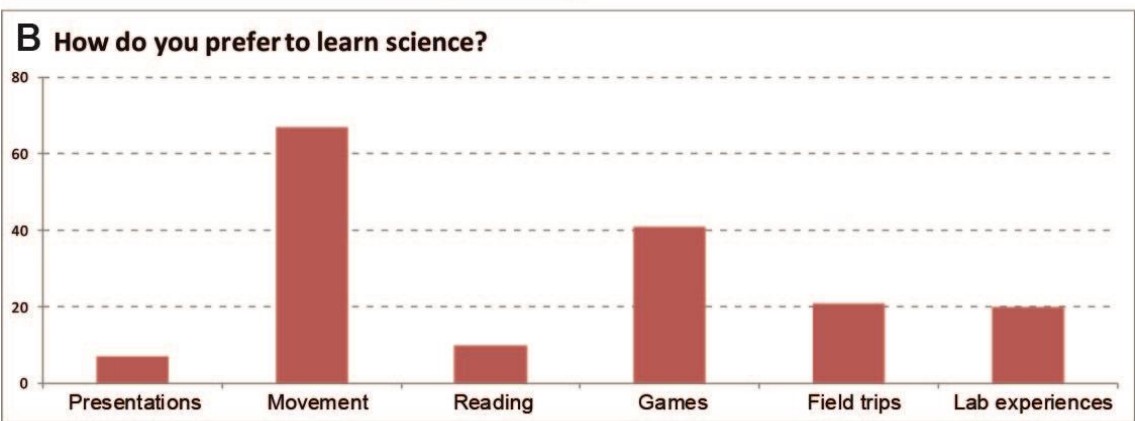

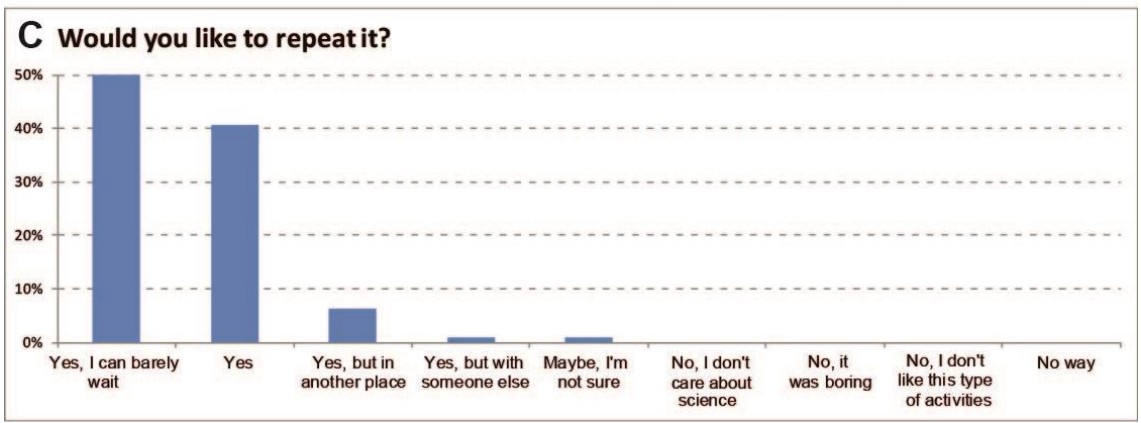


**Figure 5 - Results of inquiries for some of the questions.**
**Note: for the question about how they prefer to lean about science, multiple responses were allowed, and the**
**vertical axis is the number of responses, not a percentage.**

**4. Innovation, insights, and limitations of the interdisciplinary fusion**

The observations made throughout the activities showed that the developed and performed
activity has pros and cons in relation to more traditional forms of informal education.
The main hypothetical risks associated with the methodology application are: the detachment of
children of the activity; the disinterest of children in the scientific subject; the lack of
understanding of children about the message; shame feeling during the dance exercises; and the
little time for reflection they had to consolidate the scientific contents. Some of these risks could
not be directly observed and measured with the results of inquiries. The size of the sample (six
sessions, 112 students) was considered sufficient for a pilot test, attesting its feasibility, age
adequacy, content relevance, teachers' interest and acceptance. However, the sample size and
composition were insufficient to analyze other factors. Comprehensive analysis and conclusions
would require a comparison between the impact of this activity and another science
communication format covering the same scientific topics and age group. The lack of an
evaluation plan was the main shortcoming of this work.
The main opportunity associated with the methodology application is the engagement of
children about science concepts, by focusing attention (demonstrated by Kim (2012) as the first
step towards engagement) on the affective domain of learning, showing emotions through
movement. Furthermore, it may have the capacity to promote ocean literacy. Nevertheless, a
measurable assessment in future implementations and studies will be crucial in order to validate
the impact of such methods. The innovation of the presented activity is the enlargement of the
science communication strategies, whereby scientists communicate also through creative
dancing.
Insights from the activity development and performance can be summarized as follows:

- The interdisciplinary solution seems to be adequate as a general approach to solving
  complex issues; the complex issue here being a generalized disconnection between
  students and geosciences. The appeal to conceptual understanding, rather than
  memorization in geosciences (e.g. names of minerals and rocks, types of volcanism and
  their location, names of geomorphological features) aligns with the most necessary
  improvements in curricular guidelines identified by Afonso et al. (2013) for Portuguese
  education of sciences. The storytelling technique of contents sequencing versus a plain
  sequence of contents look as a successful technique of engagement with the activity.
- The emotional involvement in the presence of music seems to effectively encourage
  engagement, participation and willingness to take part in different experiences. Several
  positive emotions and feelings were promoted during the activity, evolving from

| 382 | anticipation, pleasure, surprise, enjoyment, to excitement, and then serenity and |
| 383 | relaxation. The assessment of emotional states was based on local observations by the |
| 384 | persons conducting and assisting/observing, both directly and by revising photos and |
| 385 | videos. Observation notes included the record of facial expressions, silence/talk/laugh, |
| 386 | and body language (heads follow/not follow the person explaining, readiness/delayed |
| 387 | movement, take a peek/indifference, jumping and frenzy in anticipation/apathy, inertia |
| 388 | or yawn). It seems fair to suppose that the pleasant memories of the playful visits to the |
| 389 | beach evoked during the activity (vacations, playing, and freedom) became also |
| 390 | associated to science and learning. The movement and improvisation is effective in |
| 391 | creativity stimulation, self-expression and stress release, thus being aligned with the |
| 392 | 21$^{st}$-century educational orientations (as demonstrated by Cone and Cone (2012)). |
| 393 | Moreover, the activity is innovative, yet not supported by screens. During the early |
| 394 | stages of the activity, shyer children tended to be reluctant to participate, very self- |
| 395 | conscious and consequently their movements are small. As the activity advanced, they |
| 396 | became more open and engaged with the proposed exercises. |

- The activity was able to mitigate some student's exclusion factors. Inclusion of students
- with diverse and special needs in the classroom has been a major focus in education
- over the past 30 years (Villanueva et al., 2012). The children's layout in space (spread
- or in two lines facing each-other), participating in chain sequencing, allows students
- with some degree of impairment to engage in the activity. Additionally, the organization
- of the activity for school classes, rather than an activity for families, assures the
- presence of children that would not participate otherwise.

- The social benefits from this type of activity can potentially include team building and
- students learn self-discipline, gain an appreciation to other movement styles, and
- discover the value of individual differences through creative exploration and problem
- solving. Socially, children enjoy interacting with others through movement (Cone and
- Cone, 2012). They laugh and talk with each other while sharing an experience that is
- fun and rewarding. The use of free (not choreographed) movements and balls can break
- the stereotype of "dancing is for girls" thus promoting gender equality. These are values
- identified in creative dance (e.g., Landalf (1997), Carline (2011), Cone and Cone
- (2012)) that can be incorporated into science communication.

- A thorough evaluation of science communication initiatives is essential to enable the
- identification of whether long-term objectives are being met, it can help to make the
- iteration of science communication initiatives more efficient, and can also highlight
- areas that need further strengthening (Illingworth, 2017). There was anecdotal evidence
- of increased familiarity and comfort with geosciences (e.g., use of scientific
- terminology by students towards the end of the activity, processes introduced by

researchers in the exposition scenes were translated to actions by students on the climax

scene), which may have been the result of the brief explanation in the beginning of the

section, reinforced by the physical exercises. In this study, due to the sporadic nature of

the event, within a major event, it would be difficult to establish a baseline of children's

knowledge prior to the intervention. After this session, the same students were involved

in a science club devoted to topics of coastal geosciences, where experiences and a field

trip were made.

• In future activities such as European Research Night 2020 and following, an improved

programme should incorporate an assessment of the students' interest and

understanding of the scientific subject, in comparison to other methods. This entails the

development and testing of a specific impact assessment design. A future evaluation

plan can include: 1) Pre-activity data on knowledge of coastal morphodynamics, this

may be done prior to the activity or be included interactively in the introductory section

by asking for experiences of waves/shorelines; 2) Pre-activity data on how pupils prefer

to learn science, and on how students with special needs interact with other students; 3)

Pre- and post-data on science capital of the teachers and pupils; 4) Teachers' and

outside observers' evaluation of emotional states during the activity; 5) Evaluation of

impacts on the researchers and creative partners; 6) Follow up data on the students'

understanding and retention of the principles being communicated at e.g. 14 days or

other time period as deemed suitable post-event; 7) Follow up with teachers in order to

assess the impact of the activity on team building, self-discipline, and appreciation for

each other's differences. At first, qualitative methods may be used to identify what

outcomes are emerging; later quantitative methods may be used to measure the strength

of the outcome, or what proportion of participants experience the different outcomes

(Grant, 2011).

• This activity was a first step towards the setting of transdisciplinary activities in

geosciences, that can meet a rather difficult balance between scientific accuracy,

stimulation of creativity, art & science bonding, integration of body-mind principles,

and promotion of inclusion of students with special needs.


**5. Final remarks**
A science communication activity for primary-grade children, was described and qualitatively
evaluated. It combines coastal science concepts, with storytelling, and creative dance
techniques. The way scientific concepts were translated into the dance class structure were
described thoroughly, to allow science communicators the chance to look behind-the-scenes of
dance creative.
The dance ability to directly improve overall learning skills (which is at least questionable,
according to Keinänen et al. (2000)) was not the purpose here. The proposal was to use art
(dance to exemplify) as a means to promote science engagement through emotional
involvement, creativity and sensory stimulation. The presence and acknowledgement of
emotions is a further way that the practice of science communication can overflow expectations
and models of it, and something else that it would be valuable to notice more in science
communicators analysis (Davies and Horst, 2016).
The proposed activity had the ability to promote social inclusion of children with special needs,
physical impairment, and kinaesthetic learners. The theme of social inclusion in the science
communication field is not new; the political value of science communication was explicit in
many cornerstones of the history of this field (Massarani and Merzagora, 2014). Nevertheless,
the exclusion from science communication activities is not only a statistical fact, but also a
neglected matter on communication research (Dawson, 2018).
Regarding the activity impacts, inquiry results showed that all children enjoyed themselves.
Nevertheless, the improvement of geoscience literacy was not measured. Yet, science
communication paradigms have shifted from science literacy (the 'deficit model') to "Science
and Society" (e.g. Bauer (2008)). This activity is aligned with the most recent paradigms, where
communication is interactive and constructive, with emphasis on dialogue, deliberation,
participation, and empowerment (Davies and Horst, 2016). It may contribute to the students
"science capital" (as defined by Archer et al. (2015)) on the dimensions: Science-related
attitudes, values and dispositions - because science was approached in an enjoyable and
engaging way, with potential to have increased openness to geosciences; Knowing people in
science-related jobs - because both people conducting the activity were researchers and were
introduced that way at the beginning, Making science relevant to the everyday lives of students
- because geoscience study objects are part of students' lives as coastal inhabitants, very
familiar with barrier islands; besides the potential for increased science literacy (evidenced by
the use of scientific terminology towards the end of the activity).
Increased science capital or science literacy by this activity are suppositions based on qualitative
observations and suppositions; an effort to a more evidence-based science communication
approach (Jensen and Gerber, 2020) is needed and is a shortcoming of this pilot work.
The addressed geoscience topics and adopted art forms can be combined in a number of ways:
for example, we can foresee as adequate, innovative and engaging, volcanology and music (e.g.,
types of volcanoes and volcanic rocks can be approached by percussion instruments and
rhythms); climate change and drama (e.g., impacts of heat waves can inspire a play); and
oceanography and poetry (e.g., waves and currents around the world can inspire poems). An
existing case of geoscience and art is the work of the artist Laura Moriarty (see
http://www.lauramoriarty.com/) who combined plate tectonics and sculpture (faults and bedding
planes approached and appreciated as blocks of a sculpture). This almost endless number of
mishmashes, on top of the aesthetical value of earth-science objects, from a desert landscape, to
a mineral, a geyser, satellite imagery, a canyon, a rocky shore, just to name a few, is an asset
worthy of further exploration in science communication of STEAM.

**Competing interests.**
The authors declare that they have no conflict of interest.

**Acknowledgements**
This study was supported by EVREST project, PTDC/MAR-EST/1031/2014, A. Matias was
supported by Investigator Programme, IF/00354/2012, and A.R. Carrasco and A.A. Ramos were
supported by FCT under the contracts DL 57/2016/CP1361/CT0002 and DL
57/2016/CP1432/CT0001, respectively. The authors are thankful for the two reviewers'
comments and contributions, in particular to Reviewer 2 that proposed a future evaluation plan
for the activity.

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
