# Peer review of "Engaging children in geosciences through storytelling and creative dance"

_Geoscience Communication, 2019_

## Referee Comment (RC1) · Anonymous Referee #1 · 26 Nov 2019

The manuscript presents an innovative learning activity for children centered on coastal geology and and highlights the benefits of interdisciplinarity in the educational context.

I think it is well with it of being shared with the Geoscience Education community, and that Geoscience Communications is the appropriate outlet to do so.

Please find below a few suggestions for further improvement of the manuscript.

Figure 1: From reading the text as well as looking at Fig. 2, my understanding was that Geology, Art, and Storytelling came together in the Science Communication activity. If this is the case, I might like to see Fig. 1 rearranged accordingly, with Science Communication at the center as the result of interdisciplinary research

References: It might be useful to add a couple of references regarding coastal geology

scientific concepts

Line 180: What is psychomotricity equipment? Could you give an example?

The first paragraph of Section 3.1 is very important, as it provides the rationale for the activity development. I suggest moving it to the introduction.

Lines 353 and 390: The existence of "kinaesthetic learners" has been long dismissed as a myth. Please remove this.

Line 363: This is currently unsubstantiated. Please provide evidence or avoid speculation.

Line 405: Those are fantastic examples. Are those your ideas (in which case you should explicitly mention that's the case) or have those been carried out before (if so, please provide references).

---

## Author Comment (AC1) · 18 Dec 2019

gc-2019-21 Reply to RC1 comments on "Engaging children in geosciences through storytelling and creative dance" by Ana Matias, A. Rita Carrasco, Ana A. Ramos, Rita Borges

"Reviewer comments" and authors' reply.

"Figure 1: From reading the text as well as looking at Fig. 2, my understanding was that Geology, Art, and Storytelling came together in the Science Communication activity. If this is the case, I might like to see Fig. 1 rearranged accordingly, with Science Communication at the center as the result of interdisciplinary research." Reply: The reviewer suggestion makes sense. We will change the figure accordingly.

[Figure]

"References: It might be useful to add a couple of references regarding coastal geology scientific concepts" Reply: Noted. We will add references.

"Line 180: What is psychomotricity equipment? Could you give an example?" Reply: Noted. We will add a brief explanation and examples.

"The first paragraph of Section 3.1 is very important, as it provides the rationale for the activity development. I suggest moving it to the introduction." Reply: Noted. We will move the paragraph to introduction.

"Lines 353 and 390: The existence of "kinaesthetic learners" has been long dismissed as a myth. Please remove this." Reply: Noted. We will remove.

"Line 363: This is currently unsubstantiated. Please provide evidence or avoid speculation." Reply: Noted. We will remove.

"Line 405: Those are fantastic examples. Are those your ideas (in which case you should explicitly mention that's the case) or have those been carried out before (if so, please provide references)." Reply: These examples are our ideas, except the example of [plate tectonics and sculpture], which is referred to the artist that developed it. To avoid misinterpretation, the sentence will be rearranged.

---

## Referee Comment (RC2) · Anonymous Referee #2 · 9 Jan 2020

The manuscript describes the development, delivery and evaluation of an activity designed to raise emotional connectivity to geoscience communication and informal education. The stated objectives are to

'. . .engage children in geosciences. . .', and to '. . .provide arguments about the importance of arts (dance) and communication techniques (storytelling) in engagement and effectiveness of geoscience programmes and develop [the audience's] willingness to participate in similar activities.'

The activity described is a thoughtful, relevant, and innovative interpretation of the underlying scientific principles. There is evidence the children enjoyed participation and that post-event many would participate in dance-based learning activities in the future. While the majority of the learning experience was based on dance, this was still

rooted in a brief dissemination style explanation at the start of the lesson plan followed by the reinforcement in the activity.

While I admire the approach to activity development I find it problematic to separate the learning activity from the communication of the science as stated in text beginning on lines 55 and 395. If the goal is to raise science capital rather than geoscience literacy, more evidence would need to be presented regarding the positive linking of the physical activity to the understanding, familiarity, and comfort with the science, or any science, being communicated. Without this, the evidence is only convincing in demonstrating an increase in social capital, at best cultural capital.

The manuscript would be greatly improved with some simple changes to the evaluation including, 1)Pre-activity data on knowledge of coastal morphodynamics – this need not have detracted from the activity as could have been included interactively in the introductory section by asking for experiences of waves/shorelines. 2)Pre-activity data on how pupils prefer to learn science: this would have greatly strengthened assertions that the activity was a preferred method rather than relying on feedback post event collected by those delivering, which has a strong likelihood to create audience bias through wanting to please the activity deliverers. 3)Follow up data on the pupils' understanding and retention of the principles being communicated at 14 days or other time period as deemed suitable post event. 4)Pre and post data on science capital of the teachers and pupils. 5)Evaluation of any impact on the researchers and creative partners. 6)Follow up with teachers on the impact of the activity on team building, etc, would be a useful metric as well.

I do however appreciate there are difficulties in collecting some of this data. It might have helped to have more teacher involvement in developing the activity to support follow up evaluation. I also appreciate and fully agree with the authors' insight into the limitations of this study (text starting line 315, and 369) and believe that careful evaluation planning integrated into the delivery would have in fact provided the data required to greatly strengthen the manuscript. The data collected could be considered

Interactive
comment

a baseline for further delivery at, for example, European Researchers' Night 2020. Finally, I fully agree with the authors' point, starting line 385, that more analysis on the emotional connection with learning is a factor that should be recognized and measured more in science communication.

Line 180: what is 'psychomotricty? Please define.

Line 215: I would like to see a reference for both Laban's theory of movement and the adaptations from Anne Green Gilbert.

Line 248: A reference or link to the EVREST project would be useful

Line 272: typo: 'brief' should be 'briefed'

Line 321: typo: 'trough' should be 'through'

Line 322: Please rephrase 'it seems to promote ocean literacy' (perhaps to it 'may have the capacity to promote…'), or present evidence that this is the case, qualitative or quantitative from pupils directly or from teachers.

Line 337: please provide your evidence, even if it is observation based, on how you assessed the presence of the 'positive emotions'.

Line 339: While the association of pleasant memories to science seems probable, I can't see the evidence presented that this is the case. Please make it clear if this is evidence based or a supposition.

Line 356: While social benefits again seem probable, I can't see the evidence presented that this is the case. Please make it clear if this is evidence based or a supposition.

Line: 367: word omission: please place 'of' between 'identification' and 'whether'.

---

## Author Comment (AC2) · 30 Jan 2020

gc-2019-21 Reply to RC2 comments on "Engaging children in geosciences through storytelling and creative dance" by Ana Matias, A. Rita Carrasco, Ana A. Ramos, Rita Borges

"Reviewer comments" and authors' reply.

"While I admire the approach to activity development I find it problematic to separate the learning activity from the communication of the science as stated in text beginning on lines 55 and 395. If the goal is to raise science capital rather than geoscience literacy, more evidence would need to be presented regarding the positive linking of the physical activity to the understanding, familiarity, and comfort with the science, or

any science, being communicated. Without this, the evidence is only convincing in demonstrating an increase in social capital, at best cultural capital." REPLY: In line 55 we state: "here (...) engagement is a loosely term referring to behaviours that demonstrate interest in, or interaction with science-related activity or experience". In line 395 we state: "Regarding the activity impacts, inquiry results showed that all children enjoyed themselves. Nevertheless, the improvement of geoscience literacy was not measured." Our point is that the physical activity contributed to the familiarity and comfort with science. We are not sure if we understand entirely the reviewer's point; nevertheless, we can add after line 395, that we have qualitative evidence of increased familiarity and comfort with geosciences (e.g., use of scientific terminology by students towards the end of the activity), which is the result of both the brief explanation in the beginning of the section, reinforced by the physical exercises. In future works, effects should be evaluated separately.

"The manuscript would be greatly improved with some simple changes to the evaluation including, 1) Pre-activity data on knowledge of coastal morphodynamics – this need not have detracted from the activity as could have been included interactively in the introductory section by asking for experiences of waves/shorelines. 2) Pre-activity data on how pupils prefer to learn science: this would have greatly strengthened assertions that the activity was a preferred method rather than relying on feedback post event collected by those delivering, which has a strong likelihood to create audience bias through wanting to please the activity deliverers. 3) Follow up data on the pupils' understanding and retention of the principles being communicated at 14 days or other time period as deemed suitable post event. 4) Pre and post data on science capital of the teachers and pupils. 5) Evaluation of any impact on the researchers and creative partners. 6) Follow up with teachers on the impact of the activity on team building, etc, would be a useful metric as well. I do however appreciate there are difficulties in collecting some of this data. It might have helped to have more teacher involvement in developing the activity to support follow up evaluation." REPLY: The reviewer suggestion is pertinent and we acknowledge the concern. We were not sure, at the beginning,

about the feasibility and receptivity of students and teachers to this activity. The activity implementation, rendered in this manuscript, demonstrated that coastal geosciences are suitable for this type of Science & Art approach and age group; and this was our main drive for trying to disseminate it within the science communication community. On the discussion and conclusion sections of the manuscript, we will reinforce that evaluation is a shortcoming of the work that needs to be acknowledged in future studies and that the step forward to scientifically demonstrate impacts (at least on the short-to medium-term) is to implement an evaluation plan, which can follow the phases proposed by the reviewer and include a methodology to evaluate separately the impact of the introductory phase from the physical activity. We gratefully thank the reviewer's generosity in taking the effort to propose a plan.

"I also appreciate and fully agree with the authors' insight into the limitations of this study (text starting line 315, and 369) and believe that careful evaluation planning integrated into the delivery would have in fact provided the data required to greatly strengthen the manuscript. The data collected could be considered a baseline for further delivery at, for example, European Researchers' Night 2020." REPLY: We are totally in agreement with the reviewer. The proposed plan (reviewer previous comment) will be included on the manuscript, in discussion section as an issue to move this (and similar) activities further.

"Finally, I fully agree with the authors' point, starting line 385, that more analysis on the emotional connection with learning is a factor that should be recognized and measured more in science communication." REPLY: We thank this remark. Furthermore, we will add a note about this analysis on the emotional connection on the evaluation plan proposed by the reviewer (see previous comments and replies) that will be added on the manuscript, at the end of the discussion section (after line 374).

"Line 180: what is 'psychomotricty? Please define." REPLY: Noted. We will add a brief explanation and examples.

"Line 215: I would like to see a reference for both Laban's theory of movement and the adaptations from Anne Green Gilbert." REPLY: We will add a brief reference to Laban's theory of movement and referred adaptations.

"Line 248: A reference or link to the EVREST project would be useful." REPLY: Noted. We will add.

"Line 272: typo: 'brief' should be 'briefed'" REPLY: Noted. We will correct.

"Line 321: typo: 'trough' should be 'through'" REPLY: Noted. We will correct.

"Line 322: Please rephrase 'it seems to promote ocean literacy' (perhaps to it 'may have the capacity to promote. . .'), or present evidence that this is the case, qualitative or quantitative from pupils directly or from teachers." REPLY: Noted. We will change as suggested.

"Line 337: please provide your evidence, even if it is observation based, on how you assessed the presence of the 'positive emotions'." REPLY: We will add elements based on our notes, videos and photographs observations.

"Line 339: While the association of pleasant memories to science seems probable, I can't see the evidence presented that this is the case. Please make it clear if this is evidence based or a supposition." REPLY: We will change to clarify that it is a supposition.

"Line 356: While social benefits again seem probable, I can't see the evidence presented that this is the case. Please make it clear if this is evidence based or a supposition." REPLY: We will change to clarify that it is a supposition.

"Line: 367: word omission: please place 'of' between 'identification' and 'whether'." REPLY: Noted. We will correct.
* * *

---

## Author Response (AR1)

**gc-2019-21**

**Reply to reviewers and editor comments on "Engaging children in geosciences through storytelling and creative dance" by Ana Matias, A. Rita Carrasco, Ana A. Ramos, Rita Borges**

"*Reviewer/editor comments*" and authors' reply. All line numbering refers to the revised version of the manuscript.

**Reviewer#1**

"*Figure 1: From reading the text as well as looking at Fig. 2, my understanding was that Geology, Art, and Storytelling came together in the Science Communication activity. If this is the case, I might like to see Fig. 1 rearranged accordingly, with Science Communication at the center as the result of interdisciplinary research.*"

Reply: The reviewer suggestion makes sense. Figure 1 was changed accordingly.

"*References: It might be useful to add a couple of references regarding coastal geology scientific concepts*"

Reply: A paragraph will basic scientific concepts on coastal geology was added (Lines 163-182).

"*Line 180: What is psychomotricity equipment? Could you give an example?*"

Reply: A brief explanation and examples were added (Lines 225-234).

"*The first paragraph of Section 3.1 is very important, as it provides the rationale for the activity development. I suggest moving it to the introduction.*"

Reply: The paragraph was moved to the introduction (Lines 130-138).

"*Lines 353 and 390: The existence of "kinaesthetic learners" has been long dismissed as a myth. Please remove this.*"

Reply: Removed.

"*Line 363: This is currently unsubstantiated. Please provide evidence or avoid speculation.*"

Reply: Removed.

*"Line 405: Those are fantastic examples. Are those your ideas (in which case you should explicitly mention that's the case) or have those been carried out before (if so, please provide references)."*

Reply: These examples are our ideas, except the example of plate tectonics/sculpture, which is referred to the artist that developed it. To avoid misinterpretation, the sentence was rearranged (Lines 518-526).

**Reviewer#2**

*"While I admire the approach to activity development I find it problematic to separate the learning activity from the communication of the science as stated in text beginning on lines 55 and 395. If the goal is to raise science capital rather than geoscience literacy, more evidence would need to be presented regarding the positive linking of the physical activity to the understanding, familiarity, and comfort with the science, or any science, being communicated. Without this, the evidence is only convincing in demonstrating an increase in social capital, at best cultural capital."*

Reply: In line 55 of previously submitted version we stated: "here (…) engagement is a loosely term referring to behaviours that demonstrate interest in, or interaction with science-related activity or experience". In line 395 of previously submitted version we stated: "Regarding the activity impacts, inquiry results showed that all children enjoyed themselves. Nevertheless, the improvement of geoscience literacy was not measured." Our point is that the physical activity contributed to the familiarity and comfort with science. We are not sure if we understand entirely the reviewer's point; nevertheless, we added in lines 412-415 that we have qualitative evidence of increased familiarity and comfort with geosciences, which is the result of both the brief explanation in the beginning of the section, reinforced by the physical exercises. In future works, effects should be evaluated separately. In lines 507-513 we elaborate on how we think science capital may have increased.

*"The manuscript would be greatly improved with some simple changes to the evaluation including, 1) Pre-activity data on knowledge of coastal morphodynamics – this need not have detracted from the activity as could have been included interactively in the introductory section by asking for experiences of waves/shorelines. 2) Pre-activity data on how pupils prefer to learn science: this would have greatly strengthened assertions that the activity was a preferred method rather than relying on feedback post event collected by those delivering, which has a strong likelihood to create audience bias through wanting to please the activity deliverers. 3) Follow up data on the pupils' understanding and retention of the principles being communicated at 14 days or other time period as deemed suitable post event. 4) Pre and post data on science capital of the teachers and pupils. 5) Evaluation of any impact on the researchers and creative partners. 6) Follow up with teachers on the impact of the activity on team building, etc, would be a useful metric as well. I do however appreciate there are difficulties in collecting some of this data. It might have helped to have more teacher involvement in developing the activity to support follow up evaluation."*

Reply: The reviewer suggestion is pertinent and we acknowledge the concern. We were not sure, at the beginning, about the feasibility and receptivity of students and teachers to this activity. The activity implementation, rendered in this manuscript, demonstrated that coastal geosciences are suitable for this type of Science & Art approach and age group; and this was our main drive for trying to disseminate it within the science communication community. On the discussion (Lines 387-388) and final remarks (Lines 514-517) sections of the manuscript, we reinforced that evaluation was a shortcoming of the work that needs to be acknowledged in future studies and that the step forward to scientifically demonstrate impacts (at least on the short- to medium-term) is to implement an evaluation plan, which followed the reviewer suggestion (Lines 461-475). We gratefully thank the reviewer's generosity in taking the effort to propose a plan.

*"I also appreciate and fully agree with the authors' insight into the limitations of this study (text starting line 315, and 369) and believe that careful evaluation planning integrated into the delivery would have in fact provided the data required to greatly strengthen the manuscript. The data collected could be considered a baseline for further delivery at, for example, European Researchers' Night 2020."*

Reply: We are totally in agreement with the reviewer. The proposed plan (reviewer previous comment) was included on the discussion section (Lines 461-475) as an issue to move this (and similar) activities further.

*"Finally, I fully agree with the authors' point, starting line 385, that more analysis on the emotional connection with learning is a factor that should be recognized and measured more in science communication."*

Reply: We thank this remark. Furthermore, we added information about emotional connection on the manuscript (Lines 410-415, 436-438, 467-468), besides references already in the previous version (Lines 488-493).

*"Line 180: what is 'psychomotricty? Please define."*

Reply: We added a brief explanation and examples (Lines 225-234).

*"Line 215: I would like to see a reference for both Laban's theory of movement and the adaptations from Anne Green Gilbert."*

Reply: We added a brief reference to Laban's theory of movement and referred adaptations (Lines 272-287).

*"Line 248: A reference or link to the EVREST project would be useful."*

Reply: The reference was added (Lines 132-135).

*"Line 272: typo: 'brief' should be 'briefed'"*

Reply: Correct (Line 340).

*"Line 321: typo: 'trough' should be 'through'"*

Reply: Correct (Line 391).

*"Line 322: Please rephrase 'it seems to promote ocean literacy' (perhaps to it 'may have the capacity to promote. . .'), or present evidence that this is the case, qualitative or quantitative from pupils directly or from teachers."*

Reply: Changed as suggested (Line 392).

*"Line 337: please provide your evidence, even if it is observation based, on how you assessed the presence of the 'positive emotions'."*

Reply: These elements were added (Lines 410-415).

*"Line 339: While the association of pleasant memories to science seems probable, I can't see the evidence presented that this is the case. Please make it clear if this is evidence based or a supposition."*

Reply: Changed to clarify that it is a supposition (Line 415).

*"Line 356: While social benefits again seem probable, I can't see the evidence presented that this is the case. Please make it clear if this is evidence based or a supposition."*

Reply: Changed to clarify that it is a supposition (Line 433).

*"Line: 367: word omission: please place 'of' between 'identification' and 'whether'."*

Reply: Correct (Line 446).

**Editor**

*"Thank you for submitting your manuscript to Geoscience Communication, and for engaging so thoroughly in the peer-review process. The activity that you report on and the context that you provide is very useful to the wider geoscience community, and would be very well suited for publication in Geoscience Communication.*

*I am recommended that this manuscript requires major revisions before publication, because whilst I think most of the issues can be addressed in a relatively straightforward manner, I am in strong agreement with Reviewer 2 that you need to carefully reconsider your evaluation*

*strategy. Whilst it is obviously not possible to get pre-workshop data now that the activity has taken place, it should still be possible to follow up with the participants to see if there has been any long-term learning as a result of your intervention. If this is not possible, then you will have to better explain limitations of your study and use this to reframe the paper. For example, are you really raising science capital, and if so then how?"*

Reply: We appreciate the editor's decision regarding our manuscript. The editor concern, referring to Reviewer #2 comment, is pertinent. As we refer above on our reply to Reviewer #2, the manuscript intention was to demonstrate that coastal geosciences are suitable for this type of Science (Coastal geoscience) & Art (Dance) approach and age group. This was our main drive to submit it to *Geoscience Communication* journal. However, we do agree that evaluation is a key topic, and that asking for students' opinion at the end of the activity is not enough. Thus, on the discussion (Lines 387-388) and final remarks (Lines 514-517) sections of the manuscript, we reinforced that evaluation was a shortcoming of the work that needs to be acknowledged in future studies and we suggest an evaluation plan, which followed the reviewer suggestion (Lines 461-475).

*"Furthermore, in Section 5 you state that the activity was 'qualitatively evaluated', but I see no real evidence of this. Figure 5 shows that the participants enjoyed the session, but this is a basic quantitative evaluation. Do you have any comments or opinions from the participants, or could you get these now? If so then these would form the basis of a very useful qualitative evaluation."*

Reply: As stated above, we agree that the qualitative evaluation of students' enjoyment and engagement, reinforced on the manuscript (Lines 410-415, 436-438, 467-468) was insufficient. Accordingly, an evaluation plan for future activities was proposed following Reviewer #2 suggestions (Lines 461-475).

Finally, please conduct a very thorough proofread of the manuscript, as there are several typographical and grammatical errors that need correcting, some of which have been picked up by the reviewers.

Reply: A thorough revision was conducted, hopefully picking all typos. The ones identified by reviewers are pinpointed above.

[revised manuscript text omitted]

---

## Author Response (AR2)

gc-2019-21

**Reply to editor comments on the revised manuscript "Engaging children in geosciences through storytelling and creative dance" by Ana Matias, A. Rita Carrasco, Ana A. Ramos, Rita Borges**

"*Editor comments*" and authors' reply.

"*1. Please revisit the quality of your figures, and ensure that they are in high resolution with well-focussed text that is easy to read.*"

Reply: We revisited all figures and we believe they are now suitable for publication. High resolution files (jpeg or tiff) are ready, if requested.

"*2. Please re-word Section 5, as you have not 'qualitatively evaluated' the activity, and indeed you have pointed out that the lack of such an evaluation is a shortcoming of this work. This is still a valuable piece of research, and is an excellent example of a pilot programme that will lead to further work and subsequent evaluation, but at the moment it would be stretching the point to say that such an evaluation has been conducted.*"

Reply: We have made changes to section 5 in order to make sure it is not explicit or implicit that we have conducted an appropriate quantitative or qualitative evaluation.

[revised manuscript text omitted]

